# Preparation, Properties and Potential of Carrageenan-Based Hard Capsules for Replacing Gelatine: A Review

**DOI:** 10.3390/polym13162666

**Published:** 2021-08-10

**Authors:** Muhammad Al Rizqi Dharma Fauzi, Pratiwi Pudjiastuti, Arief Cahyo Wibowo, Esti Hendradi

**Affiliations:** 1Department of Chemistry, Faculty of Science and Technology, Universitas Airlangga, Surabaya 60115, Indonesia; contact.alrizqi@gmail.com; 2Faculty of Advanced Technology and Multidiscipline, Universitas Airlangga, Surabaya 60115, Indonesia; arief.cahyo.w@stmm.unair.ac.id; 3Department of Pharmaceutical Science, Faculty of Pharmacy, Universitas Airlangga, Surabaya 60115, Indonesia; esti-h@ff.unair.ac.id

**Keywords:** drug-delivery system, carrageenan, hard capsule, seaweed, product innovation

## Abstract

Intense efforts to develop alternative materials for gelatine as a drug-delivery system are progressing at a high rate. Some of the materials developed are hard capsules made from alginate, carrageenan, hypromellose and cellulose. However, there are still some disadvantages that must be minimised or eliminated for future use in drug-delivery systems. This review attempts to review the preparation and potential of seaweed-based, specifically carrageenan, hard capsules, summarise their properties and highlight their potential as an optional main component of hard capsules in a drug-delivery system. The characterisation methods reviewed were dimensional analysis, water and ash content, microbial activity, viscosity analysis, mechanical analysis, scanning electron microscopy, swelling degree analysis, gel permeation chromatography, Fourier-transform infrared spectroscopy and thermal analysis. The release kinetics of the capsule is highlighted as well. This review is expected to provide insights for new researchers developing innovative products from carrageenan-based hard capsules, which will support the development goals of the industry.

## 1. Introduction

Solid dosage forms, such as soft and hard capsules, are the most widely used delivery methods for oral administration of active pharmaceutical ingredients to patients [1] because they offer better protection against oxygen, moisture and light until the drug is released. Hard capsules are produced without the addition of a non-volatile plasticiser and have been used as drug-delivery carriers for powders, granules and pellets [2]. Soft capsules, on the other hand, are produced with the addition of a plasticiser and other minor components, such as dyes and opaquing agents, and can be delivered orally, vaginally or rectally in different forms [3].

To this day, gelatine, produced from porcines or bovines [4], is still the primary ingredient in capsules, which is a concern for some religious societies [5]. Commercialisation of such capsules requires a bovine spongiform encephalopathy-free certificate due to mad cow disease [6,7]. Therefore, an alternative material for hard-capsule production is urgently needed.

The alternative materials developed for hard capsules include alginate [8], carrageenan [9], hypromellose (HPMC) [6] and strong cellulosic fibre from different plant materials such as *Dracanea reflexa* [10] and *Tridax procumbens* [11]. However, there are still some disadvantages that must be improved for future use, such as the need for alginate hard capsules to be crosslinked with another agent to form a rigid gel [8]. The disintegration process of carrageenan and HPMC hard capsules takes longer (25.79 ± 2.92 min) than conventional hard capsules (20 min) at pH 4.5 [10,12,13]. There are several factors that influence polymer disintegration time. For example, possessing parallel conformations of a polymer network increases the stability of the polymer itself, and hence, makes it more resistant to disintegration. Long polymer chains of carrageenan potentially can be broken down into oligomers [14], which can be used for tumour treatment [15].

In this paper, we review methods for the preparation of seaweed, in particular for carrageenan-based hard capsules, and summarise their chemical properties and disintegration and dissolution profiles, including the release kinetics of the capsule. We focused on carrageenan because it has more potential than HPMC or alginate for the preparation of hard capsules. The oligomerisation process in carrageenan has not been researched sufficiently to identify opportunities for further investigation. Through this review, we hope to provide sufficient background for new researchers to further develop carrageenan-based hard capsules.

## 2. Carrageenan: Potential for Fast Drug Delivery

Carrageenan is used as a gelling agent in products, such as frozen foods, jellies and yogurt [16]. Commercially, carrageenan has been produced in six different types based on their structures (Figure 1) [17]. Among the six polymers, κ-carrageenan is the most produced due to its high gelling ability caused by the C4 conformation on the 3,6-anhydro-D-galactopyranosyl that forms a helix-like structure [18]. The formation of the helix structure is supported by the enormous number of –OH groups that form many hydrogen bonds [19].

To produce hard capsules, the material needs to have high viscosity to support the formation of a gel that can be dried into a film [20]. Since the target is the human body, the recommended solvent for the production is deionised water [13]. Previous papers reported that most carrageenan-based hard capsules exhibit slow disintegration rates, and yet some drugs must deliver a drug to the target site in our bodies in <15 min [21]. Therefore, the polymer chains of carrageenan should be reduced to increase the disintegration rate.

The molecular weight (MW) of a polymer is influenced by disintegration and dissolution processes. The high MWs of polymeric composites cause longer disintegration and dissolution times, so these polymers must be hydrolysed to oligomers to reduce these times. The accuracy of MWs of polymers used in capsules and the types of drugs encapsulated require additional disintegration and dissolution studies.

Oligomers have fewer repetitive units [2,3,4,5,6,7,8,9,10,11,12,13,14,15,16,17,18,19,20,21,22,23,24,25] and lower molecular masses than polymers. Oligomer production is useful for understanding the mechanism of polymerisation, to optimise polymer production and provide a better understanding of complex macromolecules [22]. Naturally-occurring oligomers can be found in the structures of enzymes or proteins with the formation of disulphide bridges [23] and in the bacterial polyhydroxyalkanoates [24]. Synthetic oligomerisation can be performed by changing certain physical conditions, such as temperature [25], or by chemical modification (Figure 2) [26].

Carrageenan oligomers can be produced from oligomerisation of the monomers [23] or from controlled degradation of the polymeric form [27]. Degradation of a polymer to an oligomer is affected by the homogeneity level of the polymer, chemical groups on the monomers and its degree of crystallinity [27]. For example, carrageenan can be degraded to form oligomers or monomers by acid hydrolysis under a controlled pH and temperature of the solution [15,28]. There are few articles about the development of oligomers to increase the rate of dissolution in drug-delivery systems. However, we found that carrageenan has good potential for development as an oligomer-based drug-delivery system as it is easily hydrolysed by a weak acid, such as citrate buffer [13]. Accordingly, the oligomerisation of carrageenan should be performed at a controlled temperature and time, i.e., at 70 °C for 5 h [9], to form a homogeneous solution and control the oligomers formed. One purpose of this oligomerisation process is to control the rate of disintegration of carrageenan-based hard capsules. In this manner, various oral drugs can be delivered optimally in a drug-delivery system.

## 3. Hard Capsules: Preparation and Characterisation

Characterisations will help in understanding the properties of prepared hard capsules. The properties could be compared with those of standard hard capsules to determine if the developed capsules have good potential for use in a drug-delivery system. A good-quality hard capsule should comply with international standards. For example, the largest dimension of a tablet or capsule should not exceed 22 mm [29]. When a hard capsule is developed, it is necessary to ensure that it has no adverse effects, has a suitable dissolution rate, is biocompatible and safe and has the desired efficacy of the active ingredients. In line with this, the International Pharmacopoeia recommends compatibility analysis of hard capsules consisting of a uniformity test of mass and content, a viscosity test, a disintegration test and a dissolution test [30]. Development of a preparation method and characterisation results from the series of tests above should validate the compatibility of the developed hard capsules.

### 3.1. Methods of Preparation

Hard capsules, such as HPMC- [4], carrageenan- [13] and gelatine-based capsules, are typically prepared with the dipping method [20]. Prior to the fabrication process, evaluating the viscosity and gelation temperature of the various blends of materials is crucial to find the best composition [31]. There are six major steps in dipping methods: dipping, spinning, drying, stripping, trimming and joining (Figure 3), among which the most important step that determines the properties of the resulting hard capsule is the drying process. The interaction within the polymer used for the hard capsule strongly affects its drying time, which depends on the capsule’s target rigidity [32]. For example, the stronger hydrogen-bonded water of a starch-based capsule than that of an HPMC capsule will need a longer drying process. On the other hand, Ye et al. used low temperature and a short oven time (33 °C for 3 h) for drying carrageenan-based capsules due to weaker hydrogen bonds [33].

The industrial dipping process is performed by dipping pairs of stainless-steel pins into the dipping solution to form the caps and bodies. The spinning process is performed by rotating the pins to distribute the solution uniformly and avoid the formation of gas bubbles in the solution. The hard capsules are formed by drying the solution with a blast of cool air for a certain length of time. The stripping and trimming processes are performed to produce good-quality capsules. Finally, all bodies and caps are joined together into hard capsules [20].

Similar steps were also adopted at the laboratory scale [7,9,13]. Bae et al. [34] prepared a pharmaceutical starch-based hard capsule by dipping preheated pins into the warm starch solution in which the starch thermally gelled on the surface of the pins. The pins, onto which films of gelled starch solution remained, were withdrawn, followed by drying in a controlled temperature and humidity chamber for 24 h. The dried capsule pieces were then stripped, cut into various sizes and fitted together. This dipping method can be used to prepare carrageenan-based hard capsules with an adjustment in temperature and heating time. The conditions suggested are at 70 °C for 5 h [9], as mentioned earlier. The as-prepared capsule is characterised to compare its properties with those of conventional hard capsules to determine its potential as an alternative drug-delivery system.

### 3.2. Dimension Analysis

Standard dimensional requirements for hard gelatine capsules used by capsule industries are shown in Table 1, Table 2, Table 3, Table 4, Table 5 and Table 6. Based on Ridgway [20], the size of a capsule is classified into five different classes, with size 00 as the biggest and size 3 as the smallest. Kumar et al. [35] developed an osmotically controlled release system of phenylephrine hydrochloride in an ethyl cellulose capsule. Their results showed that the lengths of the cap and body of the capsule were 10.45 ± 0.34 mm and 17.42 ± 0.22 mm, respectively. The diameters of both segments were 7.21 ± 0.17 mm and 6.79 ± 0.24 mm, respectively. If we take a closer look at Table 2 and Table 6, the resulting dimensions do not comply with the mentioned standard, which implies that the dimensional analysis of an underdeveloped capsule is optional. The quality of the dipping pen further affects the resulting capsule size. Fauzi et al. [13] prepared a carrageenan-based hard capsule with diameters of 7.18 ± 0.12 mm for the cap and 7.37 ± 0.13 mm for the body. This result is also inconsistent with the standard requirement. Therefore, the dimensional specification of the standard hard capsule should not be the primary indicator of the quality of an underdeveloped capsule, and additional, comprehensive characterisations should be performed to evaluate its overall quality. We suggest that one of the ways to comply with the standard is to use an industrial standard machine to produce the capsule to achieve precise and uniform capsules. The material composites should have good gelling strength and be dried easily. Capsule production requires trial and error until the desired properties are achieved.

### 3.3. Water and Ash Content

Based on (USP) Pharmacopoeia <731>, hard capsules commonly have a water content of 10%–15%. A thermogravimetric analysis (TGA) method can be used to determine the water content of hard capsules. A typical TGA method uses 1–2 g of samples (four or more capsules) crushed into a smaller size of about 2 mm. The samples are then loaded into an oven and heated to ±2 °C of their melting point. The weight after heating is measured after 1–2 h at a specified temperature, from which the water content can be deducted [36].

USP <281> is used to analyse the ash content of hard capsules as sulphated ash. The same amount of sample is also used to analyse the ash content of hard capsules. The samples are ignited at 600 °C ± 50 °C for 30 min, cooled in a desiccator and finally weighed [37]. The water and ash content of standard gelatine hard capsules are 10.5% ± 1.5% and 1.5% ± 0.5%, respectively [38], whereas a water content of 6%–7% is reported for HPMC hard capsules [30]. Water and ash contents have not been reported so far for carrageenan-based hard capsules.

### 3.4. Microbial Limit Test

As an edible agent, it is crucial that a drug-delivery system not contain any harmful microorganisms. Therefore, USP <61> suggests a microbial enumeration test to examine the microbiological content in a sample. The microorganisms that can be detected by using this method are *Staphylococcus aureus*, *Pseudomonas aeruginosa*, *Bacillus subtilis*, *Candida albicans* and *Aspergillus niger* [39]. Analysis of various samples in the United States has shown contents of 16.98% for *Micrococcaceae* in empty hard gelatine capsules and about 24.53% for *Bacillaceae* in finished hard gelatine-capsule products (Table 7) [40]. It is also known that pure HPMC film does not exhibit any microbial activity except in films that contain food preservatives [41]. Table 7 summarises the properties of carrageenan, carrageenan-based blends, HPMC and gelatine, which show the virtues and potential of carrageenan to replace gelatine capsules.

### 3.5. Viscosity

Polymers similar to carrageenan can form gel solutions with a high viscosity index. To measure its viscosity, ASTM D3616-95 suggests that the sample be mixed in a suitable solvent for 16 to 20 h to produce a sol. The sol is removed from the mixture, and the viscosity of the solution is determined. By using this method, the swelling index of the gel can also be determined [58].

Viscosity analysis helps to predict the compactibility of a molecule. For example, a highly soluble arabinogalactan gives a low-viscosity solution due to the relatively small hydrodynamic branched structure [59]. In the production of semi-refined carrageenan, viscosity is one of the main criteria for producing the best quality product. It is also known that viscosity can be affected by the temperature and acidity of the medium [42].

Adam et al. stated that a solution with a viscosity >600 cP would support the formation of hard capsules by the dipping method [4] because it is essential for the polymer to adhere to the pins during the drying process and produce a uniform capsule with an appropriate shape [60]. This viscosity property is also useful in the development of crosslinked hard capsules in which crosslinking reduces the solubility of the capsules and extends the drug-release time [61].

It was reported that the optimum viscosity of semi-refined carrageenan is 1291.84 cP at 80 °C and heated for 30 min [42]. Compared with a gelatine solution with a viscosity of 25.60 cP at 25.5 °C [44], a carrageenan-based solution has good potential for use as a drug-delivery system. Another candidate is HPMC, with its hydrophilic nature that helps form a gel with very low viscosities of ≤100.00 cP (Table 7) [43], which enables immediate release from the pins in the dipping method [62].

### 3.6. Swelling Degree (SD)

Swelling is a kinetic process of mass transport and mechanical deformation governed by the interaction between the polymer network and the solvent [63]. In SD analysis, a capsule is soaked in 100 mL of the medium at 37 °C ± 0.5 °C. The samples’ weights were measured during the hydrolysis process in which the chemical bonding between particles of the samples is degraded, and then the particles are surrounded by the solvent particles [13,64,65,66,67,68]. On the other hand, dissolution is a process in which the solute disperses in a solvent at the molecular level.

When a hydrophilic polymer is in contact with water, penetration of the water into the polymer occurs through a diffusion process. Penetration of water causes the polymer to swell, and some of the polymer particles will then be reduced in size (chemical degradation) that eventually leads to the full dissolution of the polymer. The times required for different polymer particles to dissolve in water are known as dissolution kinetics. The process of drug dissolution and release from the polymer is known as kinetic release [69,70].

This analysis could give a good indication of whether or not seaweed-based hard capsules have better durability to dissolution in water than other types. There is no standard for SD in the production of hard capsules that we are aware of to date. Fauzi et al. found that the maximum SD of κ-carrageenan-based hard capsules (529.23% ± 128.10%) was larger than that of gelatine (145.50% + 86.04%) (Table 7), which indicated that the disintegration rate was higher for κ-carrageenan-based hard capsule than for gelatine because of its ability to inhibit the penetration of the solvent [1]. The SD of a material can be expressed by dividing the difference between the mass of the material (*m_f_*) after soaking and the mass before soaking by its initial mass (*m*_0_) [71].
(1)$$SD=\frac{m_{f}-m_{0}}{m_{0}}$$

Distantina et al. reported the SD of κ-carrageenan crosslinked with glutaraldehyde film and found that the κ-carrageenan film attained equilibrium swelling in water at about 30 min. When glutaraldehyde was added to crosslink the material, the equilibrium time was decreased significantly; wherein the crosslinked film might absorb water without dissolution [71]. Another study conducted by Estrada et al. showed that the SD for a combination of a multi-walled carbon nanotube and a κ-carrageenan hydrogel was lower than that of a blank κ-carrageenan hydrogel. This result was probably due to the reinforcement of the hydrogel structure promoted by the nanotubes that led to the formation of a tighter gel network, which lowered the SD [67].

Analysis of SD can be performed to determine the swelling kinetics of materials, a key analysis in the characterisation of crosslinked material [72]. For example, Aydinoğlu [73] investigated the swelling kinetics of novel poly(acrylamide-co-acrylic acid) hydrogels with spirulina and found that spirulina had a strong influence on the swelling kinetics due to its interaction with the acrylic acid units that influenced the acidity of the medium. In addition, SD analysis can also be related to the diffusion mechanism in which a material will be dissolved by the end of the swelling process [74].

### 3.7. Mechanical Properties

Tensile properties indicate how the material reacts to forces under tension. Tensile tests are used to determine the modulus of elasticity, elastic limit, elongation, proportional limit, tensile strength, yield point, yield strength, work of rupture and many other useful tensile properties [75]. According to ASTM D638, the test can be performed by applying a tensile force to a sample specimen and measuring various properties of the specimen under stress [76].

The test is performed by mounting the specimen in the form of a film in a machine and subjecting it to tension. The tensile force is recorded as a function of increased gauge length (Figure 4). When a material reaches its flexibility limit, it will no longer be able to return to its normal shape; i.e., it undergoes deformation, which can be seen in an A–B pathway (Figure 4) [77].

Generally, the tensile properties of a polymer can be improved by adding a crosslinking agent to a polymer matrix [78]. Alvarado et al. investigated the tensile strength of a composite film made of chitosan, fish gelatine and microbial transglutaminase and concluded that the tensile strength decreased with an increase in gelatine content. This result was caused by the degree of deacetylation that affected the physical (e.g., tensile strength), chemical and biological properties of chitosan [79]. In the carrageenan-based film development, it was found that tensile strength values were higher for κ-carrageenan films (39.34 ± 0.51 MPa) than for gelatine (31.03 ± 0.74 MPa) [45] and HPMC (19.90 ± 1.20 MPa) films (Table 7) [34]. This finding suggests that the hard capsules prepared from carrageenan would be stronger than those from gelatine and HPMC.

Good mechanical properties assure better quality control in capsule manufacturing, i.e., by making it easier to produce a uniform weight and prevent oxidation or hydrolysis, leading to poor stability. Moreover, in the quality control process of a product, the capsules with optimum tensile strength will have good flexibility [80,81].

### 3.8. Surface Morphology Analysis Using Scanning Electron Microscopy (SEM)

SEM has been used worldwide in many disciplines and is recognised as an effective method for image analysis of organic and inorganic materials on a nanometer to micrometre scale. SEM works at a magnification scale of up to 50,000× [82] and even 1,000,000× in the latest models to produce extremely detailed images of a wide range of materials [83].

The scanning was performed by using high voltage (1.0–25 kV) to accelerate the secondary electrons between the anode and cathode. This process produces an enlarged image of the subject’s surface. The magnification is shown by a ‘times’ symbol (X); e.g., 1000× means 1000-fold magnification [84]. To prepare the material to be analysed by SEM, it can be prepared as a film [4,13,85]. The preparation of a film depends on the physical properties of the material itself. For example, Li et al. applied a solution of a mixture of pectin–chitosan complex plasticised by sorbitol onto an acrylic glass plate and dried at 50 °C to make a film [86].

As reported by Fauzi et al. [13], both carrageenan and carrageenan, crosslinked with maltodextrin and plasticised by sorbitol films, exhibited invisible pores even at 5000× magnification. This result supported what was observed by Krόl et al. [46]. A carrageenan film showcased pores observed on its surface at the imaging scale of 200 nm, whereas no pores were observed on the surface of gelatine film at that scale. Thus, carrageenan pores are bigger than gelatine pores. Pure HPMC film has a smooth surface, which may imply a less ductile film, as well as a homogeneous and uniform matrix with no pores observed at the scale of 30 μm (Table 7) [41].

The presence of pores and their sizes may be related to the matrix mechanical properties. For example, the presence of a crosslinking agent would decrease the size of pores and hence make the matrix stiffer, whereas adding a plasticiser would achieve the opposite [67]. In addition, these interactions should be confirmed by Fourier-transform infrared (FTIR) spectroscopy from observation of new bond formation due to crosslinking or emergence of new peak(s) from plasticiser functional group(s) [87]. By controlling the pore size of a matrix, the rate of diffusion of the solvent into the matrix could be affected as well [1]. This way, the disintegration rate of a hard capsule could be predicted on the basis of the SEM analysis of the matrix surface. Other aspects should also be considered in determining the disintegration rate, such as the SD. As mentioned above, the SD of carrageenan-based hard capsules is larger than that of gelatine-based capsules, making the disintegration rate of the former faster [13].

### 3.9. Molecular Weight (MW)

Using more modern analytical methods, such as gel permeation chromatography (GPC), polymer MWs of 10,000 g·mol^−1^ to 400,000 g·mol^−1^ can be determined fairly accurately [88]. Examples of high MW bio-polymers include gum Arabic [4], gelatine [88], lignin [89] and carrageenan [90,91]. An example of MW determination of carrageenan was reported by Uno et al., who found that the number average MWs of these carrageenans ranged from 193 kDa to 324 kDa [47]. It is also known that the MWs of HPMCs ranged from 10 kDa to 22 kDa, depending on the percentage of methyl and hydroxypropyl substitutions [48]. Compared with gelatine, with MWs ranging from 7.1 kDa to 131.6 kDa [49] and HPMC, carrageenans have more potential for controlled modification of their structure and MW through the formation of pre-designed oligomers to develop a better drug-delivery system (Table 7).

GPC is also useful for dissolution studies of polymers in an organic solvent. This technique can help assess the degree of polymerisation and the number of monomer subunits that a polymer contains [92]. Therefore, this technique was used to characterise a composite of seaweed-based hard capsules with various components, such as the carrageenan-based hard capsules that Fauzi et al. [13] developed. However, the drawback of this technique is that there may be possible interactions between the sample material and column fillers that could interfere with the analysis [92].

### 3.10. Thermal Properties

Thermal characterisation, in which differential scanning calorimetry (DSC) and TGA are used in combination is an important method for studying a material’s behaviour under temperature change. DSC measures the chemical or physical transition of a material when the temperature is increased or decreased following ASTM D3418-12, in which a sample is heated or cooled under a specified purge gas at a certain flow rate. The energy changes in the material are marked by the absorption or release of energy, resulting in endothermic or exothermic peaks [93]. This process changes the state of matter, and melting and crystallisation processes are some of the important indicators in DSC and TGA. A melting point (*T*_m_) is confirmed both theoretically and experimentally to indicate at what temperature reduction in particle size and significant reduction in viscosity (and hence melting) occurs. On the other hand, the glass transition point (*T*_g_) describes the temperature at which the mechanical properties of a material change from hard and brittle to soft, deformable or rubbery [94]. In addition, a new peak in FTIR spectroscopy can represent bond formation in a crosslinked system that may result in the change in *T*_m_ and/or *T*_g_ in DSC analysis [95]. On the other hand, TGA measures the mass change of a material during a process, such as decomposition, due to the temperature change, in accordance with ASTM E1131-08 [96]. Therefore, a combination of both DSC and TGA data provides fundamental information about the thermal properties and chemical structure of a material [97].

In gelatine-capsule analysis, thermal changes are correlated with *T*_g_ to determine its hardness. It was found that water was the determining factor for the equilibrium of the gelatine network formation within a short period of time [98] and that the inhibition of water evaporation could reduce capsule damage [99]. Bigi et al. observed that the *T*_g_ of dried gelatine film occurred at 90 °C–92 °C [52], with a one-stage decomposition feature of 15% weight loss observed by TGA [100]. On the other hand, Perfetti et al. observed a much higher *T*_g_, i.e., 280 °C–300 °C, of HPMC film [51] (Table 7), expanding its thermomechanical stiffness for hard-capsule applications.

Mahmood et al. found that carrageenan experienced five stages of decomposition: at 90 °C, 192 °C, 245 °C, 350 °C and 780 °C (Table 7). The decompositions occurred in different stages due to the presence of moisture, sulphate groups and carbohydrate backbone fragmentations [50]. Kianfar et al. formulated a carrageenan-based drug-delivery system for ibuprofen, and DSC and TGA were used as some of the characterisation techniques. The TGA showed that the residual water content of the film was 5 wt.%, whereas DSC showed that the crystallisation point of ibuprofen was −53.87 °C. No reports of the *T*_g_ of carrageenan were found. These results, with other supporting analyses, indicated that carrageenan mixed with other polymers could be a potential drug-delivery system for buccal drug delivery [101]. In another reference, DSC confirmed that the presence of a heterogeneous polymer network in a carrageenan-based drug-delivery system provided a tunable diffusion rate [102]. This heterogeneity can be achieved by adding either a potassium cation for a slower diffusion rate or a sodium cation for a faster diffusion rate to the gels [103]. The aforementioned information shows that DSC and TGA are helpful for designing better drug-delivery systems.

### 3.11. Fourier-Transform Infrared Spectroscopy (FTIR)

Vibrational spectroscopy is a valuable investigative tool because it provides information about the bond formation or loss, structural rearrangements and other molecular properties of materials [104]. FTIR is useful for characterising the potential interactions in the chemical structures of capsule materials [105]. By using FTIR along with nuclear magnetic resonance, DSC [95] and near-infrared analysis, the crosslinking between polymer chains can be studied well [104].

Based on ASTM 168, a sample’s absorbance of infrared light produces a unique FTIR spectral fingerprint specific to a class of material [106]. An extracted fish gelatine in acetic acid, for example, shows three major peaks at 3600–2700 cm^−1^, 1900–900 cm^−1^ and 400–900 cm^−1^ that indicate the presence of amide groups because gelatine is essentially a protein (Table 7) [56]. For HPMC, the unique peaks observed at 1053 and 944 cm^−1^ are associated with an alkyl-substituted cyclic ring containing an ether linkage [55] (Table 7). Hard capsules made of κ-carrageenan have fingerprint region peaks at 1248, 930, 847 and 805 cm^−1^ (Table 7). When the κ-carrageenan was crosslinked with maltodextrin, the peak at 1248 cm^−1^ was broadened, which indicated a crosslink had formed [13]. In addition, Table 8 shows that the combination between two polymers could significantly change the IR band. The change can be in the form of band shifting [33,53], peak shape [13] or even new peak formation [4]. These changes depend on the way the polymers interact with each other. For example, He et al. stated that the higher the concentration of the locust bean gum blended with κ-carrageenan, the greater the shift of the O–H stretch band [33]. Thus, FTIR characterisation is useful for determining the interactions between two polymers for the development of drug-delivery carriers.

Thermal properties and SEM can also be used to confirm the formation of crosslinking, complementing FTIR results. Distantina et al. stated that the presence of glutaraldehyde as a crosslinking agent to carrageenan improved its thermal stability. The control sample showed an endothermic peak at 91 °C and an exothermic peak at 167 °C. When glutaraldehyde was added, the endothermic and exothermic peaks both increased to 96 °C and 172 °C, respectively [71]. Meng et al. showed that the presence of calcium ion, as a crosslinking agent for carrageenan, increased the surface roughness of the film, as shown in SEM images. Other effects of crosslinking include an increase in the thermal decomposition onset, as observed in TGA thermograms [107].

## 4. Disintegration Process

Complete disintegration of a capsule is defined as the state in which no residue of the unit, except fragments of insoluble coatings or capsule shells, remains on the screen of the test apparatus and is a soft mass with no palpably firm core [108]. In other words, it is a mechanical breakdown process of a material that forms smaller sizes in a solvent without changing the chemical structure of the material (Figure 5) [1]. Disintegration and swelling are two connected processes. Swelling occurs when a material is penetrated by a solvent and expands. When the swelling process reaches its maximum capacity, the material disintegrates [109].

There are at least two different definitions of disintegration that can be adopted depending on the purpose of the research. Based on the USP-32 General Chapter <2040>, the disintegration process refers to the rupture of the drug-delivery system, i.e., the opening of soft-shell capsules [110]. Another definition that is favoured recently and recommended by the European Pharmacopoeia (Ph. Eur) 21 and USP 108 is that ‘disintegrated’ means that the material needs to be completely unobserved by the unaided eye [111]. The first method will need an additional agent, such as lactose [4], to help the observation, and the disintegration is stated to begin when the release of the agent is first observed. On the other hand, the second method does not need an additional agent and will need a longer observation time since the disintegration requires that the material be completely invisible by unaided eyes [112]. Based on a comparison, it is suggested to use the full disintegration (second) method since there is no additional agent that may affect the disintegration process.

This disintegration analysis is useful to determine the quality of a drug-delivery system since the purpose of such a system is to deliver a drug into the body and release it at a certain time [21,108]. Disintegration analysis can be performed in vivo [113] or in vitro [114]. Different materials will exhibit different disintegration times. For example, HPMC hard capsules will disintegrate within 16 ± 5 min, whereas gelatine hard capsules will take 12 ± 4 min in the human body [57]. Modification of the structure of carrageenan-based hard capsules can lead to various disintegration times. For instance, carrageenan–alginate hard capsules will be disintegrated in deionised water within 12.80 ± 1.43 min, whereas carrageenan–amylum hard capsules will take 25.79 ± 2.92 min [9] and carrageenan–maltodextrin hard capsules will take 18.47 ± 0.19 min [13].

Disintegration time, a potentially major barrier in facilitating drug release, is an important property for a capsule. Disintegration time depends on the packaging materials, filling materials, preparation process, pharmaceutical excipient properties and manufacturing process of the product. If a slow disintegration time is needed, then a carrageenan–amylum formulation can be employed; otherwise, a carrageenan–alginate formulation can be used for a faster disintegration time.

## 5. Dissolution Process and Release Kinetics of Drug-delivery Systems

In 1931, Hixson and Crowell [115] developed a dissolution concept in which the surface area is equal to the mass of the material (ω). With the assumption of a constant change in concentration, the Hixson–Crowell Equation is expressed as follows:(2)$$\omega_{0}^{\frac{1}{3}}-\omega_{f}^{\frac{1}{3}}=kt$$
where *ω*_0_ and *ω_f_* are the initial mass and the mass at time *t*, respectively, and *k* is a constant.

Noyes and Whitney [116] continued in early 1990 by conducting an experiment that would be the foundation of dissolution analysis. They put a sample material in a glass cylinder and then dipped it into the water in a glass bottle. From the experiment, Noyes and Whitney derived the Noyes–Whitney Equation as follows:(3)$$\frac{dx}{dt}=C\left(S-x\right)$$
where *S* represents the solubility of the material, *x* is the concentration at time *t* and *C* is a constant.

The concept of drug release was developed by Higuchi [117,118] and is now considered to be an important parameter for determining the performance of a drug-delivery system. The drug-release kinetics connect the concepts of the diffusion process and dissolution process. The equation below expresses the Higuchi formula of drug-release kinetics:(4)$$\frac{q\left(t\right)}{q_{\infty }}=K\sqrt{t}$$
where $q_{\infty }$ is the cumulative amount of the drug released at infinite time, *q(t)* is the cumulative amount of the drug at time *t* and *K* is the Higuchi constant. This model is useful for studying the formulation of a drug-delivery system matrix.

The non-Fickian diffusion concept was initially proposed by Frisch et al. [119] in which there was a deficiency in Fick’s diffusion for a swollen polymer, also known as a glassy polymer. The concept was then developed by Ritger and Peppas [120], who developed the Peppas–Ritger equation (also known as the Power Law equation):(5)$$\frac{M_{t}}{M_{\infty }}=kt^{n}$$
where $M_{\infty }$ is the cumulative amount of the drug released at infinite time, *M_t_* is the cumulative amount of the drug at time *t* and *n* is the diffusion exponential of the drug released. Such an exponential term (*n*) can generally be used to describe the diffusion mechanism of a material (Table 9) [121].

When a drug-delivery system exhibits a non-Fickian diffusion mechanism, analysis using the Peppas–Sahlin equation [122] could be used:(6)$$\frac{M_{t}}{M_{\infty }}=k_{1}t^{n}+k_{2}t^{2n}$$
where $M_{\infty }$ is the cumulative amount of the drug released at infinite time, *M_t_* is the cumulative amount of the drug at time *t*, *n* is the diffusion exponential of the drug released, *k*_1_ is the diffusion constant and *k*_2_*t*^2*n*^ is the non-Fickian contribution caused by the relaxation process of a swollen polymer. This equation is ideal for analysing the 60% release point of a drug.

Other release-kinetics models were developed to help determine the release mechanism of a material. A zero-order model was analysed in detail by Varelas et al. [123], and the equation is expressed as follows:(7)$$Q_{1}=Q_{0}+k_{0}t$$
where *Q*_1_ is the dissolved material at time *t*, *Q*_0_ is the initial concentration of the material and *k*_0_ is the zero-order constant. This model is recommended for a transdermal matrix drug-delivery system. Another model to be mentioned is the first-order release-kinetics model [124], derived from the Noyes–Whitney equation, as shown in equation 8. *Qt* is the dissolved material at time *t*, *Q*_0_ is the initial concentration of the material and *k*_1_ is the first-order constant.
(8)$$\ln Q_{t}=\ln Q_{0}+k_{1}t$$

The mentioned equations might be used to evaluate the best release mechanism of a drug from a matrix. To ensure this further, statistical evaluation should be used, such as the Akaike Information Criterion [67]. The derived mechanism will help us to understand how a drug is released from a matrix to evaluate its potential as a drug-delivery system.

Carrageenan-based capsules have better solubility at pH 4.5 than at pH 1.2 or pH 6.8 because the citrate buffer, used as the medium that imitates human body fluid, interacts better with the polymer. The best adopted drug-release kinetic mechanism for this capsule was Peppas–Sahlin model at pH 1.2 and 4.5. The capsules are released completely in 40 min in acidic environments, indicating that the capsules have good potential for use with oral drugs [13].

## 6. Conclusions

Exploration in the development of hard capsules with the goal of replacing gelatine for drug-delivery systems is progressing. Some of the alternative materials that were studied are alginate, carrageenan, HPMC and cellulosic fibre. The production of hard capsules is performed in six major steps: dipping, spinning, drying, stripping, trimming and joining. Comparing properties of carrageenan, HPMC and gelatine, by using several characterisation methods, such as GPC, SEM, FTIR, thermal, SD analysis, mechanical analysis, viscosity analysis, disintegration, dissolution and release-kinetics analysis were presented to determine the best option for replacing gelatine as the principal constituent in a drug-delivery system. Despite being in the early developmental stage in which further investigations are needed, carrageenan-based hard capsules have properties comparable to those of gelatine and show good potential as an alternative to gelatine hard capsules due to its ability to be modified with other polymers to make a composite with the targeted properties for a better drug-delivery system. Finally, based on the research and development of carrageenan-based hard capsules our research groups have conducted, we are currently in the process of implementing scale-up to work towards semi-commercial production.

## Figures and Tables

**Figure 1 polymers-13-02666-f001:**
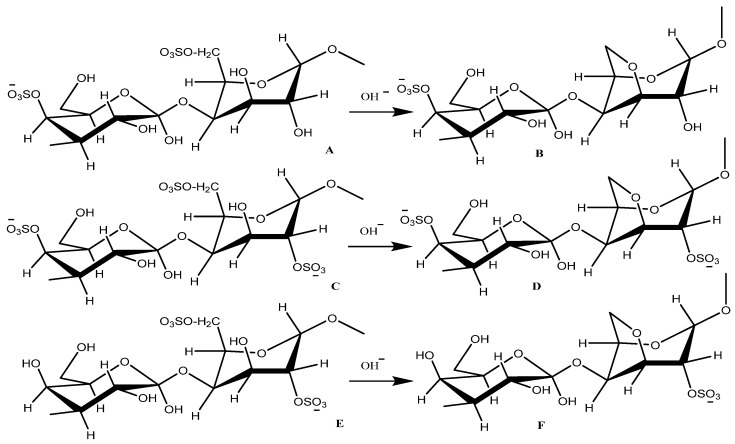
Six types of carrageenan. A = μ- carrageenan, B = κ- carrageenan, C = ν- carrageenan, D = ι- carrageenan, E = λ- carrageenan and F = θ- carrageenan.

**Figure 2 polymers-13-02666-f002:**
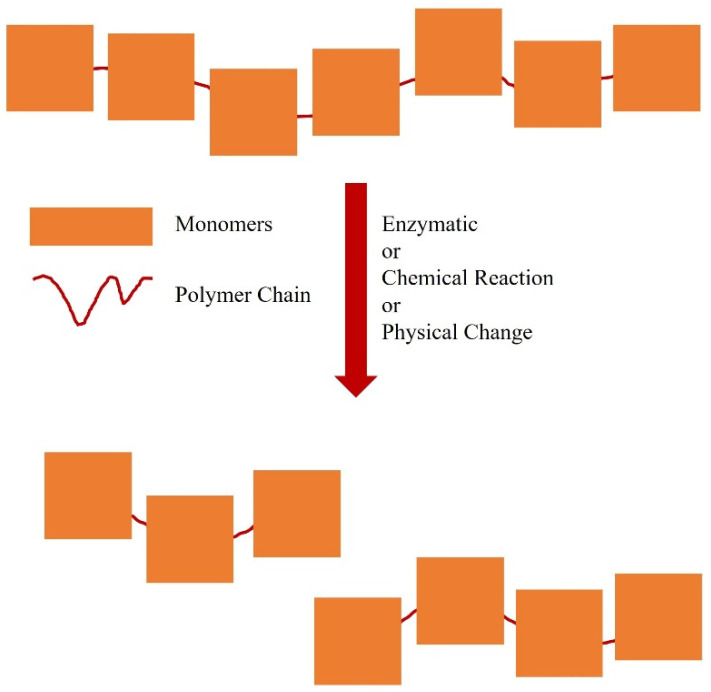
Some synthetic strategies to produce an oligomer from a polymer.

**Figure 3 polymers-13-02666-f003:**
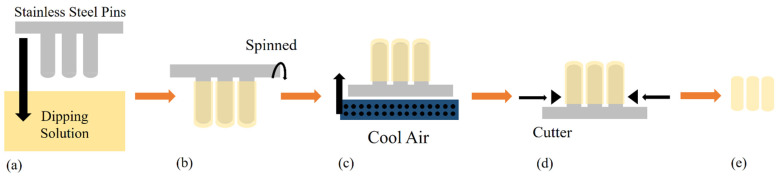
Hard-capsule production using the dipping method. (**a**) Dipping, (**b**) spinning, (**c**) blowing, (**d**) stripping and trimming and (**e**) joining.

**Figure 4 polymers-13-02666-f004:**
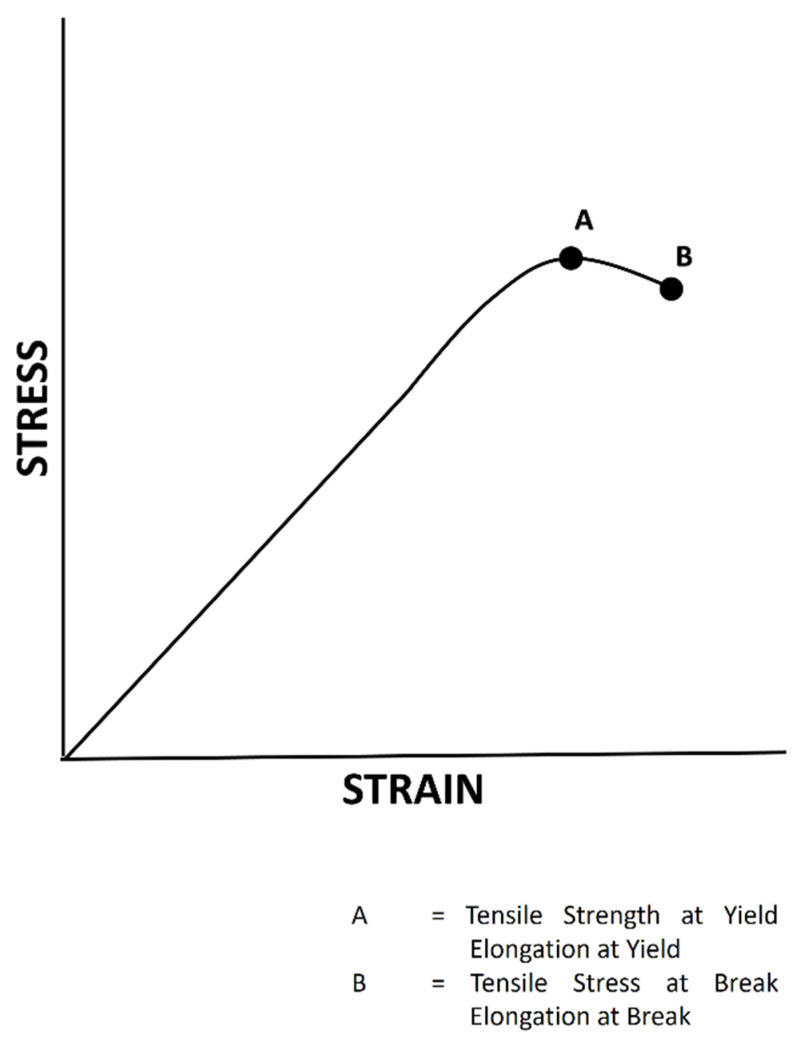
An illustration of a load–elongation curve from a tensile test.

**Figure 5 polymers-13-02666-f005:**
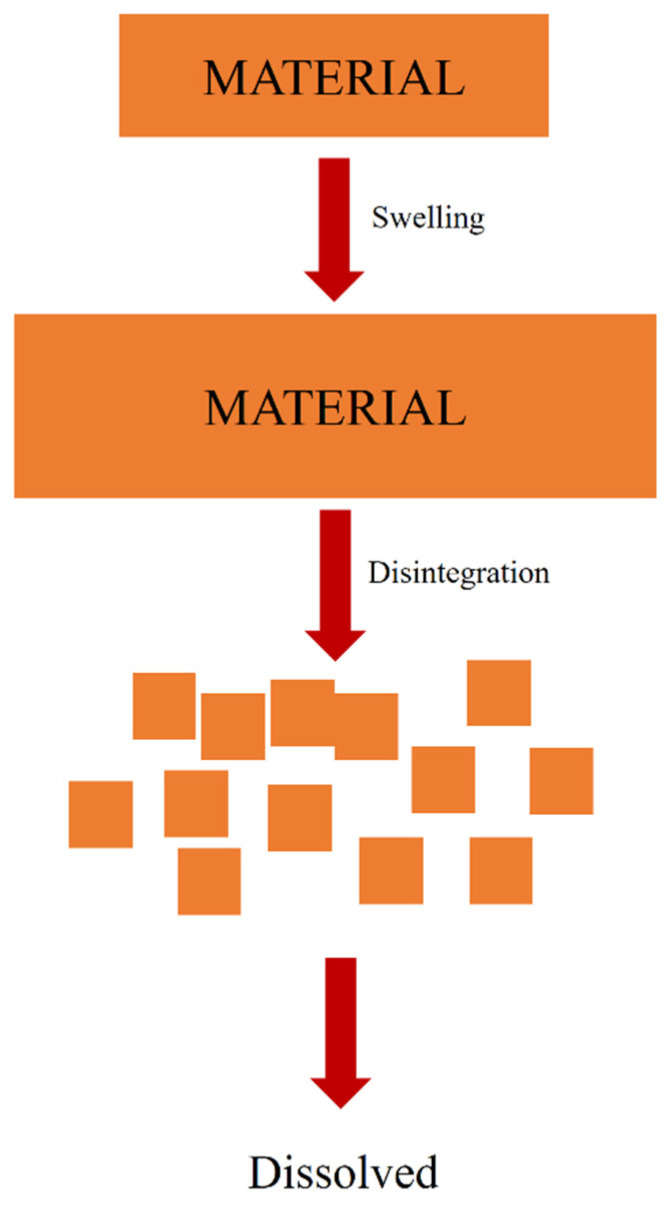
Disintegration process mechanism.

**Table 1 polymers-13-02666-t001:** Weight specification of hard gelatine capsules.

Capsule Size	Weight (mg)		
	Minimum	Capsule Size	Maximum
00	110	00	110
0	87	0	87
1	67	1	67
2	55	2	55
3	46	3	46

**Table 2 polymers-13-02666-t002:** Length specification of the segment of hard gelatine capsules.

Capsule Size	Body (mm)	Cap (mm)
00	19.50–20.50	11.50–12.50
0	17.90–28.90	10.20–11.00
1	16.00–17.00	9.300–10.30
2	14.80–15.70	8.500–9.400
3	13.25–14.05	7.600–8.500

**Table 3 polymers-13-02666-t003:** Length specification of hard gelatine capsules.

Capsule Size.	Before Locking (mm)	After Locking (mm)
00	25.00–26.00	23.30–24.45
0	23.15–23.90	21.00–22.00
1	20.45–21.20	18.90–19.70
2	18.60–19.50	17.35–18.00
3	17.20–18.10	15.50–16.70

**Table 4 polymers-13-02666-t004:** Thickness specification of hard gelatine capsules.

Capsule Size	Body (mm)	Cap (mm)
00	0.20–0.22	0.21–0.23
0	0.19–0.21	0.20–0.22
1	0.19–0.21	0.19–0.21
2	0.19–0.21	0.19–0.21
3	0.18–0.20	0.19–0.21

**Table 5 polymers-13-02666-t005:** Capacity specification of hard gelatine capsules.

Capsule Size	Capsule Volume (mL)	Weight Capacity for Powdered Drug (mg) based on the Density			
		0.6 g mL^−1^	0.8 g mL^−1^	1.0 g mL^−1^	1.2 g mL^−1^
00	0.95	570	760	950	1140
0	0.68	408	544	680	816
1	0.50	300	400	500	600
2	0.37	222	296	370	444
3	0.30	180	240	300	360

**Table 6 polymers-13-02666-t006:** Diameter specification of hard gelatine capsules.

Capsule Size	Capsule Diameter	
	Body (mm)	Cap (mm)
00	8.15 ± 0.10	8.51 ± 0.10
0	7.29 ± 0.10	7.60 ± 0.10
1	6.55 ± 0.10	6.88 ± 0.10
2	6.04 ± 0.10	6.32 ± 0.10
3	5.56 ± 0.10	5.79 ± 0.10

**Table 7 polymers-13-02666-t007:** Comparison of hard-capsule properties.

No	Property	Carrageenan	Ref.	Carrageenan-Based	Ref.	HPMC	Ref.	Gelatine	Ref.
1	Water and Ash Content	*N/A*	*-*	*N/A*	*-*	Water content: 6%–7%	[30]	Water content: 10.5% ± 1.5% Ash content: 1.5% ± 0.5%	[38]
2	Microbial Activity	*N/A*	*-*	*N/A*	*-*	HPMC film does not contain any microbial activity	[41]	16.98% *Micrococcaceae* and 24.53% *Bacillaceae*	[40]
3	Viscosity	1291.84 cP at 80 °C and cooked for 30 min	[42]	*N/A*	*-*	≤100.00 cP	[43]	25.6 cP at 25.5 °C	[44]
4	Swelling Degree	*N/A*		529.23% ± 128.10%	[13]	*N/A*	-	145.5% ± 86.04%	[13]
5	Mechanical Properties	39.34 ± 0.51 MPa	[45]	*N/A*	*-*	19.90 ± 1.20 MPa	[34]	31.03 ± 0.74 MPa	[45]
6	Surface Morphology by SEM	No pores at 5000× magnification	[13]	Pores observed on the surface at 200 nm	[46]	No pores at 30-μm scale	[41]	No pores at 200 nm scale	[46]
7	Molecular Weight	193 kDa to 324 kDa	[47]	*N/A*	*-*	10 kDa to 22 kDa	[48]	7.1 kDa to 131.6 kDa	[49]
8	Thermal Properties	Five stages decomposition: 90 °C, 192 °C, 245 °C, 350 °C and 780 °C	[50]	*N/A*	*-*	T_g_ = 280 °C–300 °C	[51]	One-stage decomposition; T_g_ = 90–92 °C	[52]
9	Fingerprint Spectrum on FTIR	1248, 930, 847 and 805 cm^−1^	[13]	Additional fingerprint peaks will be detected	[13,33,53,54]	1053 cm^−1^ and 944 cm^−1^	[55]	3600–2700 cm^−1^, 1900–900 cm^−1^ and 400–900 cm^−1^	[56]
10	Disintegration Rate	*N/A*	–	18.47 ± 0.19 min in deionised water	[13]	16 ± 5 min in human body	[57]	12 ± 4 min in the human body	[57]

*N/A = Not Available.*

**Table 8 polymers-13-02666-t008:** FTIR spectral band of seaweed-based material.

Raw Material (Cited Reference)	Modified Material (Cited Reference)
κ-Carrageenan - D-galactose-4-sulphate 842–847 cm^−1^ [13,54,59,104,105] - 3,6-anhydro-D-galactose 925 to 930 cm^−1^ [13,54,59,104,105] - C–O and C–C stretching of pyranose ring 1033- to 1038 cm^−1^ [54,59,106] - Sulphate ester 1219–1248 cm^−1^ [54,59,106] - Free OH, SO_2_ and NH groups stretching vibrations 3323 to 3331 cm^−1^ (broad) [54,59,106] ι-Carrageenan - Additional sulphate ester 805 cm^−1^ [105] - D-galactose-4-sulphate 845 cm^−1^ [105] - 3,6-anhydro-D-galactose 930 cm^−1^ [105] λ-Carrageenan - High sulphate content 820–830 cm^−1^ (broad) [105] - D-galactose-4-sulphate 845 cm^−1^ [105] - 3,6-anhydro-D-galactose 930 cm^−1^ [105]	κ-Carrageenan crosslinked with corn starch [53] - Shifted glycosidic linkages 1038 cm^−1^ to 1028 cm^−1^ - Shifted C–O bond stretching of corn starch 1150 cm^−1^ to 1152 cm^−1^ - Shifted sulphate ester 1219 cm^−1^ to 1236 cm^−1^ κ -Carrageenan blended with locust bean gum [33] - Shifted O–H stretching of hydroxyl groups because of an increase in hydrogen bond 3458 cm^−1^ to 3438 cm^−1^ κ-Carrageenan crosslinked with maltodextrin and plasticised with sorbitol [13] - Broadened sulphate ester band 1248 cm^−1^ κ-Carrageenan crosslinked with Arabic gum [54] - A new band of sulphate esters 400 cm^−1^ - A new band of glycosidic linkage 1034 cm^−1^

**Table 9 polymers-13-02666-t009:** Diffusion exponential and release mechanism of a material.

Diffusion Exponent (n)			Mechanism
Film	Cylinder	Sphere	
0.50	0.45	0.43	Fickian Diffusion
0.50 < n < 1.00	0.45 < n < 0.89	0.43 < n < 0.85	Anomalous Transport
1.00	0.89	0.85	Case-II Transport
>1.00	>0.89	>0.85	Supercase-II Transport

## Data Availability

The data presented in this study are available in the article.

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
