# Peer review of "Preparation, Properties and Potential of Carrageenan-Based Hard Capsules for Replacing Gelatine: A Review"

_polymers, 2021, doi:10.3390/polym13162666_

Round 1
Reviewer 1 Report
The authors of the work “Preparation, Properties, and Potential of Carrageenan-based Hard Capsules for Replacing Gelatin: A Review” have the intent to highlight the suitability of carrageenan as natural polymer for the production of capsules for drug delivery. The use of a seaweed-derived polymer for that purpose retains a relevant rationale in terms of physical properties, such as the possibility to tune the responsiveness/disintegration of capsules. In addition, the origin of carrageenan is important to overcome various religious inconveniences with respect to gelatin-based systems. Hence, the usefulness of this review work is relevant and proper for the scientific community in sight of real improvements in the field of drug delivery. Nonetheless, a major revision of the manuscript is required in order to properly discuss the above-mentioned rationale, highlighting the promising perspectives for the effective implementation of carrageenan.
The authors should properly answer to the following reviewer’s comments.
General comments:
1) The authors must meticulously revise the manuscript due to the presence of various errors. In fact, the reading of the work is not easy due to the presence of long and unclear sentences and typos (e.g., page 1-lines 38-39, page 2-lines 51-52, page 4-lines 129-131, page 6-line 188). The current form is generally difficult to read.
Specific comments:
1) The abstract must be more concise and based on shorter sentences (e.g., lines 16-19) to improve its effectiveness. In that regard, some repetition can be observed concerning the potential use of seaweed-based capsules (e.g., line 18 and line 20). These modifications can help the authors to show the aim of their work quickly and properly.
2) Paragraph 2 is supposed to provide the reader an optimal understanding of the suitability of carrageenan for drug delivery on the basis of its physico-chemical properties. Nonetheless, such aim is not effectively realized. The consecution of the sentences is not well-structured and hence the general meaning is not clear. The authors must revise this paragraph and organize it in a better manner. For example, the importance of polymer molecular mass on the self-organization and dissolution/disintegration process can be clearly discussed before the explanation of the usefulness of oligomerization. In fact, these aspects are highly correlated, and their relationship should be properly reported.
3) Paragraph 3.2: do the authors have an idea on how to overcome the actual limitation in terms of the dimension of capsules to meet standard requirements? After the statement at page 5-lines 167-170, a critical perspective of the authors could be added to suggest a likely improvement in that regard.
4) Paragraph 3.6: the authors should highlight and stress the differences between physical dissolution and chemical degradation (e.g., hydrolysis). Polymeric networks based on hydrophilic macromolecules are supposed to initially dissolve and then degrade in chemical terms. Moreover, since the authors considered the entire set of parameters describing the dynamic response of capsules based on polymeric networks, swelling and dissolution behavior should be discussed in the same paragraph. In fact, the overall responsiveness in contact with watery environment can be considered as a function of both swelling, dissolution and even chemical degradation, as also indicated by the authors in paragraph 4. The evident correlation between these phenomena is important to evaluate the resulting release kinetics of encapsulated drugs. Hence, these aspects should be incorporated and critically discussed within the same paragraph to facilitate the reading.
5) Paragraph 3.7: why are mechanical parameters important? The authors should discuss the importance of the mechanical parameters regarding the production of capsules. Did the authors consider the correlation between the mechanical properties and the responsiveness in contact with watery environments?
6) Paragraph 4: the authors should properly indicate the role of the formulation on the tunability of disintegration process. Only general statements are reported in the manuscript in that regard. On the basis of this improvement, the reader can generate stronger ideas for future formulations based on carrageenan.
7) Paragraph 6: the conclusive paragraph should represent a critical evaluation of the promising perspectives of carrageenan for drug delivery applications. On the basis of the state of the art, the authors should properly discuss their general ideas to suggest potentially effective studies and improvements towards the real implementation of carrageenan as principal constituent of capsules for drug delivery.
Author Response
Dear Reviewer-1
I answered the comments in separated file

Reviewer 2 Report
The papers attempts to highlight the preparation of carrageenan-based hard capsules, and summarize their chemical properties. Authors found that carrageenan has a good potential to be developed as an oligomer-based drug delivery system as it is easily hydrolized by a weak acid like citrate buffer. Comparing properties of carrageenan, HPMC and gelatin thought several characterization methods to determinate the best option to substitute gelatin as a drug delivery system. They found that carrageenan has a good potential to be developed as an oligomer-based drug delivery system as it is easily hydrolized by a weak acid like citrate buffer.
Authors provide sufficient background for new researchers to further develop carrageenan-based hard capsules.
Please, check that references are in according to author´s guidelines
Author Response
Dear Reviewer-2,
We answered the comments in separated file

Round 2
Reviewer 1 Report
Reviewer’s Comment
The authors must carefully read and revise the manuscript again, as various errors can be detected (e.g., page 4-line 137 “shorther”, page 10-line 219 “3.5. Viscority”, page 12-line 292 “Figure 4. An ilustration of load-elongation curve from a tensile test.”).
The authors should revise again paragraph 3.6 to clearly describe the processes of swelling and hence erosion of polymeric constructs, such as polymer-based capsules (previous reviewer’s comment 4). The added statements are unclear. The following works can be helpful to understand and briefly highlight these ideas in this review: https://doi.org/10.1016/0142-9612(96)85755-3 and https://doi.org/10.1002/pola.26765.
In addition, no modifications can be detected in the second part of the work. Hence, the answers of the authors to reviewer’s comments 5-7 are incomplete: added parts/revisions in the text of the manuscript are missing (no parts highlighted in yellow are present).
The work can be considered for publication after further major revisions.
Author Response
Thank you very much for Reviewer's comments
We provided a point by point in the separated pages

Round 3
Reviewer 1 Report
Reviewer’s comment
The authors added the responses to the previous reviewer’s comments. The overall quality of the work is improved in terms of content. Nonetheless, the authors must read with more attention the manuscript, as unclear sentences and statements are still present. No relevant modifications in terms of language were carried in the manuscript, thus indicating that the previous reviewer’s indications were not properly satisfied. In fact, even already suggested corrections were not accomplished in the last version of the manuscript (i.e., “Viscority”, page 10-line 217). Other poorly intelligible sentences and errors are indicated below as indicators of the poor language even after revision (round 2):
- “Having parallel conformation increase the stability of a polymer, hence more resistant to disintegration.” (Page 2-lines 48-49)
- “The energy changes in the material as compared to a reference standard is marked by absorption or release of energy, corresponds to endothermic or exothermic peak.” (Page 13-lines 367-369).
- “Carrageenan-based hard capsules also have a good potential to be developed as drug delivery systems because of its various disintegration times, depend on its structural modification.” (Page 17-lines 474-476)
Many other poorly written sentences can be found in the manuscript. This peculiarity is negatively affecting the overall quality and message of the review work. Hence, it is necessary to meticulously and precisely revise the paper in order to reach a proper language level, in accordance with the one required for the journal (Polymers, IF 4.329). A complete and precise revision of the manuscript must be carried out, as the actual quality of the work is still not sufficient for publication. In that regard, the contribution of a native speaker is strongly suggested.
Author Response
Dear Reviewer,
Thank you very much for evaluation and suggestion.
We answered point by point in the separate file.
With best regards,
Pratiwi
